# Identifying the Needs of Older Adults Associated with Daily Activities: A Qualitative Study

**DOI:** 10.3390/ijerph20054257

**Published:** 2023-02-27

**Authors:** Juan Carlos Briede-Westermeyer, Paula Görgen Radici Fraga, Mary Jane Schilling-Norman, Cristhian Pérez-Villalobos

**Affiliations:** 1Department of Design Engineering, Universidad Técnica Federico Santa María, Avda. España 1680, Valparaíso 2390123, Chile; 2Departamento de Design e Expressão Gráfica, Universidade Federal do Rio Grande do Sul, Porto Alegre 90010-150, Brazil; 3Department of Medical Education, Universidad de Concepción, Concepción 4070386, Chile

**Keywords:** specific needs, product design, older adults, solutions, multidisciplinary, healthy aging, cocreation

## Abstract

Introduction: By 2050, older adults will constitute 16% of the world population; hence, there is an urgent demand and challenge to design solutions (products and services) that meet the needs of this age group. This study sought to analyse the needs that impact the well-being of Chilean older adults and present possible solutions through the design of products. Methodology: A qualitative study was used, where focus groups were held with older adults, industrial designers, health professionals, and entrepreneurs on the needs and design of solutions for older adults. Results: A general map was obtained that linked the categories and subcategories related to the relevant needs and solutions, which were then classified in a framework. Conclusions: The resulting proposal places the needs in different fields of expertise; and thus, enables positioning, broadening, and expanding upon the map to share knowledge, between the user and key experts, to co-create solutions.

## 1. Introduction

Life expectancies are progressively rising, causing a significant demographic shift known as population aging. Globally, the number of people aged 80 years and older is expected to triple by 2050, reaching 16% of the world population [1]. This means that more people will need help to cope with age-related impairments and maintain their quality of life [2]. However, it has also been seen that despite increased life expectancy, healthy life expectancy has not increased in the same proportion, meaning that many will reach this stage with health difficulties that could impair their quality of life [3].

People in this period of life are also exposed to changes in their nervous system, cognition, and memory, as well as in the sensory level and musculoskeletal system [4], which can affect their quality of life and, subsequently, life expectancy. In this aspect, design plays an essential role in generating solutions that can respond to the needs of this age group and contribute to their well-being and quality of life.

This work is part of ambitious research whose main challenge is to co-create solutions with older adults. 

To explore co-creation with this age group, the focus first targeted self-supporting older adults; thus, reducing the complexity of both the research process (user) and compliance with ethical considerations. It is expected that, in the second stage, bedridden older adults with disabilities and a network of support persons can be incorporated.

Hence, the objective of this work, which focuses on self-sufficient older adults living in Chile, is to identify and categorize the needs of products and services directly related to aging, to support older adults in performing their daily activities. 

### 1.1. Well-Being and Quality of Life in Older Adulthood

Healthy aging is a process that involves the promotion and upkeep of functional capacities that allow for well-being in old age. This is the interrelation between a person’s intrinsic capacity (physical and mental capacities) and the environment where they live, considering their relationships with other people, participation in the community, access to services, social and health policies, and their ability to interact with the physical environment [5].

Well-being at this stage of life considers physical (maintaining a good physical level), mental, and cognitive health (self-knowledge, ability to see things in perspective, lifelong learning, and faith), social participation (social support, financial security, community participation), and independence (having the physical and mental ability to live without support, as well as being financially autonomous from family and friends) [6,7].

In addition, it has been seen that aging and living autonomously generate a sense of identity and well-being in older people, and a sense of belonging to the community. Additionally, many daily activities generate a sense of satisfaction, contributing to older adults’ well-being [8,9,10].

With this in mind, home adaptations become very important to ensure the autonomy and well-being of older adults. Likewise, social participation and support through regular contact with friends and neighbours give them a sense of support and positively influence their well-being and survival [8,10]. These matters can be solved through product and service design by creating products that allow people to remain autonomous for as long as possible.

### 1.2. Designing Solutions for Older Adults

Older adults represent a growing population segment and are an active user group participating in all aspects of life [11]. Therefore, systems (products and services) that do not consider older users’ unique needs and abilities will likely fail in their ability to support their use and adoption by this increasingly growing population segment [12]. Moreover, in some cases, failure to consider older users in the design process may result in slow and error-prone system use [12]. In this scenario, errors can have serious consequences, exposing older users to an increased risk of injury or death. 

Products and services are defined for this research as products, devices, objects, and services as well as product–service systems. This concept is a service–product combination that forms a marketable set of products and services, jointly capable of meeting a client’s needs [13]. For this research, the built environment, especially dwellings, serves as the backdrop for most of the activities performed by older adults. 

Under this proviso, unless older adults are involved in the design process, designers may incorrectly anticipate their needs and preferences [12]. Hence, it is essential to create and implement new solutions that consider the active participation of different stakeholders, experts, and future users (older adults), as they will more closely represent the needs and possible solutions, and confirm that these solutions meet the needs raised [14].

It is in this aspect that user-centred design [15,16] proposes a designer expertise approach [17], who, as a specialist, seeks to best meet the users’ needs based on research and empathy to address an iterative process; and where the requirements of the users themselves are integrated to define the end-user profile, guiding the creative process through testing and feedback [18]. 

Participatory approaches to design [19,20] present perspectives to involve end users during the creative process. This approach comes from a participatory culture, where the user is considered an expert in their own life, seeking to be actively involved as co-creators in a process to ensure that the product or service addresses their needs [17].

Despite these guidelines, sectors as diverse as architecture [21] or product design [22] have excluded groups of users, either because of their age, gender, or social status; or because they present additional challenges to designers, such as difficulties in communicating or empathizing with their life experience, due to a lack of motor, cognitive, or social skills [23]. Considering and analysing these “extreme” users [24] offers valuable learning in understanding how they have to look for creative alternatives to face and handle needs not solved by a market that has excluded them.

The involvement of older adults in the design process has had social, technical, and economic justifications. First, the barriers and challenges faced by older adults that affect their quality of life, independence, and health have been made transparent [25], while involving them in decision-making may lead to greater acceptance of the products [25] and support them in healthy aging [26]. There are also high expectations in the use of technologies for well-being [27]; in particular, from the use of robots and virtual pets that support physical and mental health, combat isolation, and improve the quality of life for this age group [26]. However, if technological barriers are not considered, these products will not thrive. This is especially salient, given that the market aimed at meeting the needs and requirements of an older population, also called the “silver economy”, could reach an estimated €6.4 trillion by 2025, equivalent to 38% of Europe’s GDP, while generating 88 million jobs [28].

### 1.3. Needs Identification

The design process, to approach the design of solutions, seeks to identify needs and consequently, develop a solution that satisfies them. This begins by identifying a user profile and their needs; the problems that emerge from unmet needs; and then, through creative stages, developing ideas and concepts for solutions.

The Center for Research and Education on Aging and Technology Enhancement (CREATE) model analysed the needs and desires of users in this age group. Through research, they identified six activity domains: health, living environment, work, and volunteer activities; leisure activities, communication and social engagement; and transportation [29]. This model was created to guide the development of products and services explicitly oriented to older adults. 

In Figure 1, the needs and desires of users can be seen, where each level of the pyramid refers to a type of need. At the base are basic activities of daily living (B-ADL), defined as “necessary for survival and a measure of autonomy,” followed by instrumental activities of daily living (I-ADL), which are “required for complete independence,”. At the top are enhanced/advanced activities of daily living (E-ADL), which are “important for a high level of life satisfaction and well-being” [29]. This model was made using the framework outlined by Czaja et al. [29] and De Vriendt et al. [30].

Considering that older adults are a fast-growing population with specific unmet needs, and using this framework as a reference, this work aimed to analyse the needs that impact the well-being of Chilean older adults and their possible solutions through the design of products and product-service systems. 

## 2. Materials and Methods

### 2.1. Study Design 

A qualitative study was made, that involved collecting and analysing non-numerical data (e.g., text, video, or audio) to understand concepts, opinions, or experiences. The advantage of this type of research is that it can be used to gather in-depth insights into a problem or generate new ideas for research with a phenomenological design [31]. The data gathered through a qualitative study reflects on how a person experiences or understands their world and showcases their particular worldview regarding the question or problem proposed by the researcher. This research method seeks to capture the participants’ perceptions and impressions, rather than find universal truths. This was important, as co-creation relies on users’ experiences and expertise to design products and services, and considers the experience that other professionals that interact with them (e.g., medical staff, among others) or who are considered in the design process (e.g., designers, entrepreneurs) have. The general methodological sequence used in the research is described in Figure 2.

### 2.2. Participants

The study had forty participants: sixteen older adults, nine industrial designers, five entrepreneurs, and ten health professionals who work with this type of user, from the cities of Santiago, Concepción, and Talcahuano. For participant selection, the inclusion criterion for professionals was to have at least 4 years of professional experience; and for older adults, to be over 60, classified as self-sufficient with 43 or more points in EFAM A [32], and have resided in the last 12 months in urban areas of the Biobío, and metropolitan regions. Participants were chosen through convenience sampling [33].

The study participants’ data, divided by focus group, are presented in Appendix A.

### 2.3. Data Collection Techniques

Focus groups were used to gain access to the participants’ narratives and experiences. In these, each participant shares their thoughts, feelings, perspectives, and perceptions about the topic presented [34,35,36], promoting a breadth of thoughts and motivation among participants [37]. This allows the researcher to understand the experience from the participant’s point of view through the verbal and nonverbal communication of data [35] arising from perceptions, interactions between participants, and the joint building of meanings [34,35,37,38]. Here, the researcher acts as a rather passive facilitator or moderator, opens the conversation, and intervenes sporadically, leaving the focus group participants to be the protagonists. 

Six two-hour focus groups were held: 2 with older adults (Concepción and Talcahuano); 2 with industrial designers (Concepción and Santiago); 1 with entrepreneurs (Concepción); and 1 with health professionals (Concepción). Each focus group had between 4 and 10 participants.

In the focus groups, the research project was presented first, and information was presented didactically, by sharing a brief PowerPoint presentation with the goals of the project and a research context so that the participants could understand it. The focus groups of designers, entrepreneurs, and health professionals explored the following topics from a script of questions (Appendix B): older adult activities, products for activities, and opportunities for co-creation. The focus groups of older adults focused on a script of questions that addressed the following topics (Appendix C): household activities performed by older adults, the importance assigned to household activities, difficulties faced by older adults, suitability of commonly used products to the needs of older adults, and the proposal of products. 

### 2.4. Procedure

Authorizations were obtained from the institutions participating in FONDECYT projects No. 11701137 and No. 1201987. Convenience sampling was used to recruit participants. Invitations were sent by e-mail to health professionals, design professionals, and entrepreneurs, while older adults were contacted by telephone. The meetings were held in a room at the Department of Medical Education in the Universidad de Concepción. In the case of industrial designers, two focus groups were held, one in Concepción (UdeC) and the other in Santiago (in a Coworking space).

Before starting the focus group, the researcher and the participants read and signed the informed consent form. Then, a list of the participants of each focus group was generated, the audio of all the sessions was recorded, and later, the transcription of each focus group was made. The anonymity of the subject was ensured in the transcription.

### 2.5. Data Analysis

For data analysis, content analysis of the focus groups was made, creating codes related to needs and solutions. The needs and solutions were analysed, grouping them and subsequently contrasting them with the needs identified in the analysed literature [29,30]. After that, concepts were linked together by affinity, and a needs/solutions map was created. This map was built considering the theoretical principles of the CREATE framework [29] and the model proposed by De Vriendt [30], but adapted and built using the codes obtained in the focus groups.

The Atlas ti version 7.5.4 software was used to code the focus groups; then, Microsoft Excel to analyse the identified concepts. After this, the Miro app was used to create the needs/solutions map.

## 3. Results

The focus groups were held in the cities of Santiago, Concepción, and Talcahuano. Participant recruitment was addressed by convenience sampling, leaving the gender variable open. A total of 16 subjects were older adults, with 9 industrial designers, 5 entrepreneurs, and 10 health professionals who work with older adults. Women represented 60%. This preponderance could be explained by their willingness to participate in social gatherings and a culture that taints everyday activities with a feminine bias. The same is true for care activities and can explain why 9 out of the 10 health professionals working with older adults were women. On the other hand, 8 out of the 9 industrial designers were men.

The key concepts (and/or key situations) from the focus group analysis, describing the needs of older adults as perceived by each focus group, are identified below. Preliminary results are described in Table 1.

### 3.1. Analysis/Extraction

Two categories were identified that describe their relationship concerning the needs of older adults: (1)Needs of older adults: derived from health professionals and older adults.(2)Solutions for older adults: derived from industrial designers and entrepreneurs.

The number of concepts per category was:-needs: 362 concepts;-solutions: 231 concepts.

The key concepts were analysed sequentially to highlight those which the two categories had in common. A new category of “common concepts” was established; and, as a result, 111 common concepts were found (Figure 3). 

These common concepts are structured in four columns to compress the information (Table 2). These concepts are general, many of which could be sub-categories of others and with different levels (general and specific).

### 3.2. Framework: Needs Categorization Proposal

Based on the analyses, a proposal for classifying concepts is presented using the relationship between the theoretical frameworks and the results obtained in this study. 

Since the purpose is to design products and/or services for older adults, the process of classifying the identified needs is conducted using the CREATE and the DeVriendt models. The latter classifies the activities and links them to the users’ human dimensions; the socio-cultural factors of the environment; the physical environment; and how these influence the specific aspects and characteristics of the older adult (demographic, psychographic, perceptual, cognitive, psychomotor), the use of technology, and the need to solve tasks (complexity, familiarity, collaboration, time demands).

To link the frameworks, first, the definitions of the six activity domains (transportation, health, etc.) were obtained using the theoretical proposals. However, two concepts were not defined: “Health” and “Work and voluntary activities”.

The bibliography behind the frameworks was reviewed to use terms already found within the study area to generate the levels of the activity domains. The books “Designing for Older Adults: Case studies, Methods, and Tools” [12], and “Designing for Older Adults: Principles and Creative Human Factors Approaches” [29], along with the article by De Vriendt et al. [30], were essential to obtain examples of activity domains and be able to generate the levels (Table 3).

### 3.3. Affinity Map

A map of the list of concepts (Figure 4) was drawn up to analyze them to identify similarities and relationships. Using the Miro platform (https://miro.com/es/ accessed on 15 September 2020), the concepts were organised as follows:Those that shared terms were grouped. For example, “physical skills”, “mental skills”, and “cognitive skills” were grouped.Those that did not share words were linked by interpreting their meaning. For example, “caution” was grouped with “safety”, “insecurity”, and “use of technology for safety”. Other concepts were also grouped, but some remained unconnected.In this way, groups of words grew; however, their relationship became ambiguous. For example, the grouping of “physical abilities”, “process of accepting physical limitations”, and “physical limitations” became distant from the “mental abilities”, “mental activities”, and “intellectual exercise” group. Therefore, it was decided to separate the groups following their meaning rather than their word relationship, creating two new groups from the previous one.The new word groups were associated under a concept that encompassed them, creating 26 clusters, such as “Safety”, “Autonomy”, “Technology”, etc. This way, concepts that were left out in the other steps were classified, leaving a smaller group unclassified.Some concepts could be in more than one cluster; for example, “redesigning products to suit user needs” could be in both the “Design Process” and “Product” categories. It was decided that it should be in both, and background colours were used to mark their overlapping. Each color only had the function of showing the overlap.As overlaps appeared, clusters were organized for those concepts that could be in more than one category. For example, the term “tourism” was at the intersection of 2 clusters: mobility and leisure activities.The clusters were organised to share at least one relationship and were all linked. Twenty-three clusters were joined, leaving out three, namely labour, retirement, and education, linked together by the concept of “retirement”. Thus, a macro group of 23 clusters and a small group of 3 clusters were obtained.Within the clusters, the closeness between concepts was communicated by linking them. This was based on the repetition of words from step 1. For example, the concepts “routine”, “active routine—daily activities”, and “routines that favour medication intake (forgetfulness)” are linked in the cluster “Routine”; however, they are separated from the concept “daily activities to maintain active cognitive skills”. Similarly, the latter is on the boundary between the “Routine” and “Mental” clusters, communicating that it is close to this concept without overlapping.Some concepts could not be catalogued in the first 3 clusters (by words, interpretation, or cluster creation) as they had distant meanings and were catalogued at the end. For example, “education” is the only concept that alludes to this concept, or “warning” as it is ambiguous.

To facilitate the legibility of the contents of Figure 4, it was subdivided into five sections (Figure A1, Figure A2, Figure A3, Figure A4 and Figure A5), which are attached individually in the Appendix D.

Finally, with the mind map ready and the terms analyzed, the matrix in Table 2 was completed, obtaining the matrix for Table 3.

The frameworks facilitated the creation of clusters and the identification of relationships between concepts. This exercise was useful strictly for those concepts that were the last to be categorized due to their ambiguity, such as “warning”, “redesign”, and “education”; however, that made sense when they were within clusters and related to other terms.

The domain “Work and volunteer activities” does not to consider the level of “Basic Activities of Daily Living “, as these are not activities needed for survival. On the other hand, the domain “Leisure Activities” is restricted to the level of “Enhanced/Advanced Activities of Daily Living”. The 111 concepts were then classified into 15 categories of the 18 available ones, and the result in Table 4 was obtained.

The concepts obtained from the focus groups generated clusters to group them by relationship. These clusters can also be categorized for better understanding. This is described in Table 5.

By organizing the numerous concepts using the theoretical frameworks model, it is possible to define which category an older adult product or service should be designed for, how to place the needs in different fields of expertise; and thus, be able to place, expand upon, and develop the map identified to share knowledge between the user and key experts, to co-create solutions.

## 4. Discussion

This study looked to identify the needs of older adults and the product design solutions that can be developed through co-creation.

The code categorization within the theoretical frameworks [29,30] already highlights similarities between the needs identified in both studies. In turn, other recurring themes are identified in studies addressing the needs of older adults.

Physical limitations are identified as a background for developing products that adapt to the specific needs of this age group based on these changes. Thus, the deterioration that could be experienced in mobility, cognitive functions, and the development of frailty [39] are also associated with a lower quality of life and fear of death [40,41]. These findings are consistent with other studies that also report these problems [6,7,42].

Furthermore, it is considered that quality of life in old age also implies social participation. However, this study, as well as others, highlights the isolation that older adults are exposed to due to a lack of support networks and opportunities for social participation in the community [40,42,43]. Likewise, it is seen that on an urban scale, the lack of pavement upkeep shows a disregard for the needs of older adults at a municipal level [42].

Work and leisure are also topics mentioned in this study, and similarities with other studies are also observed. The quality of life of older adults is affected by leaving work and the lack of space for leisure, given that life at this stage requires redefining purpose and time towards other interests and responsibilities [40,42]. A critical aspect is the economic situation of older adults, which tends to be quite precarious [43]. On the other hand, a large part of the routine is devoted to daily life activities, as well as to the family by caring for grandchildren, which gives some kind of social participation and transcendence to older people at this age [40,42,43].

One aspect that makes the context in which life develops during aging relevant is related to the person’s autonomy and the assistance and care required by others. Both in this study and others, it has been identified that autonomy is an aspect that needs to be prolonged as long as possible since it provides well-being in this period of life [8,9,10,43]. Given the universal needs identified in different contexts and the discovery in this study of 251 concepts not addressed by the solutions, the existence is highlighted of potential demands that require solutions in products and services for older adults. These values showcase the idea that greater interaction between the two categories can enable the development of products and/or services that are more closely aligned with the needs of older adults, explaining the importance of user-focused, expert-backed, co-creation projects. Furthermore, concept maps make the links between identified needs and solutions more explicit. Therefore, visually explaining the connections between concepts has the potential to assist the product development process for older adults.

### 4.1. Limitations

This study must be understood in the socio-cultural context of the participants and the reality of older adults in Chile. The methodology used to identify needs and solutions from focus groups can be replicated with other groups in different socio-cultural contexts. However, it is considered that the framework provides an important foundation that allows generalizing results, as similarities are found between the results of this research and the theoretical framework from research made in the United States [29] and Belgium [30].

### 4.2. Future Lines of Work

The mapping and findings are expected to be applied in co-creation workshops with older adults to deepen and break down the needs and solutions in a collaborative process aimed at developing a solution.

Future lines of work should also involve other groups of older adults with emerging age-related problems, and explore new techniques and considerations where the co-creation of solutions can be better conducted when working with older adults.

## 5. Conclusions

The inputs obtained in the various formats become a basis for starting the design of solutions using participatory methodologies such as co-creation, while narrowing and streamlining the intervention for relevant issues, and promoting exchange and socialization between users and stakeholders to contribute to a shared understanding of the design needs of older users.

The organizational structure that the frameworks include makes it a good instrument for starting co-creation processes regarding older people’s needs.

## Figures and Tables

**Figure 1 ijerph-20-04257-f001:**
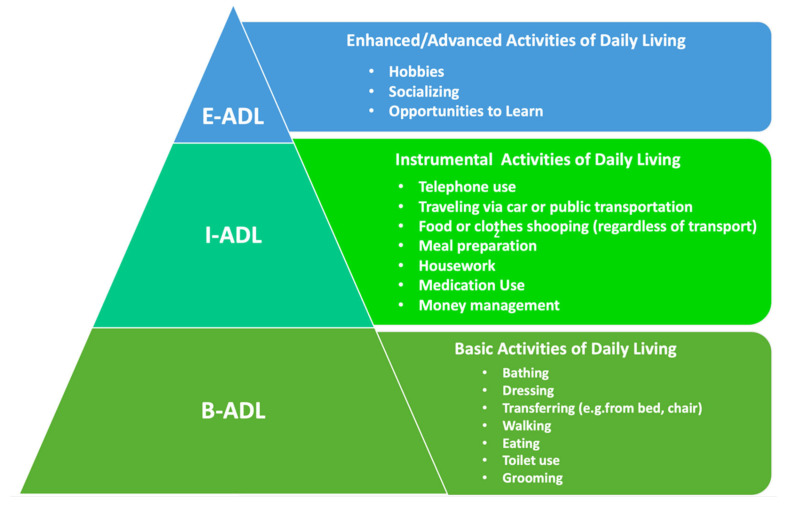
Framework adapted from Czaja et al. p. 257 [29] and De Vriendt et al. [30]. Source: developed by the authors.

**Figure 2 ijerph-20-04257-f002:**
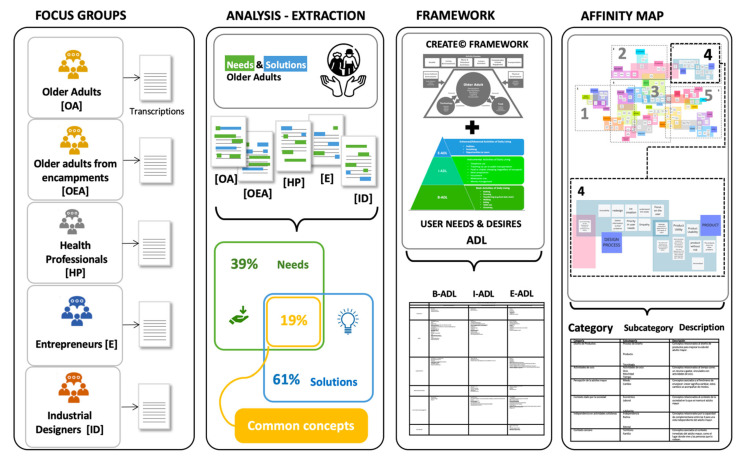
Methodological sequence addressed in the study. Source: developed by the authors.

**Figure 3 ijerph-20-04257-f003:**
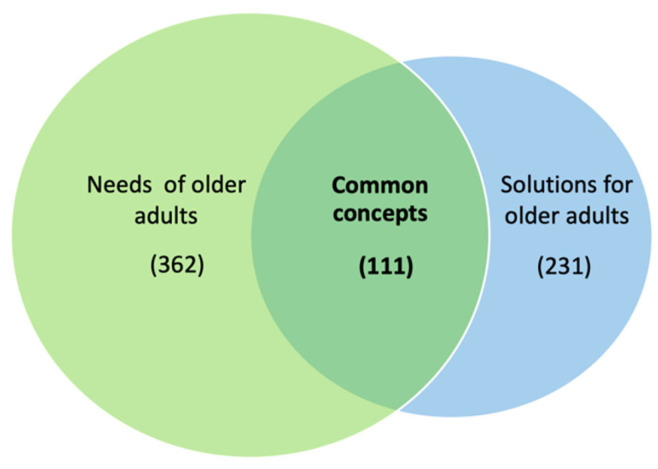
Concepts analysis chart. (Source: developed by the authors).

**Figure 4 ijerph-20-04257-f004:**
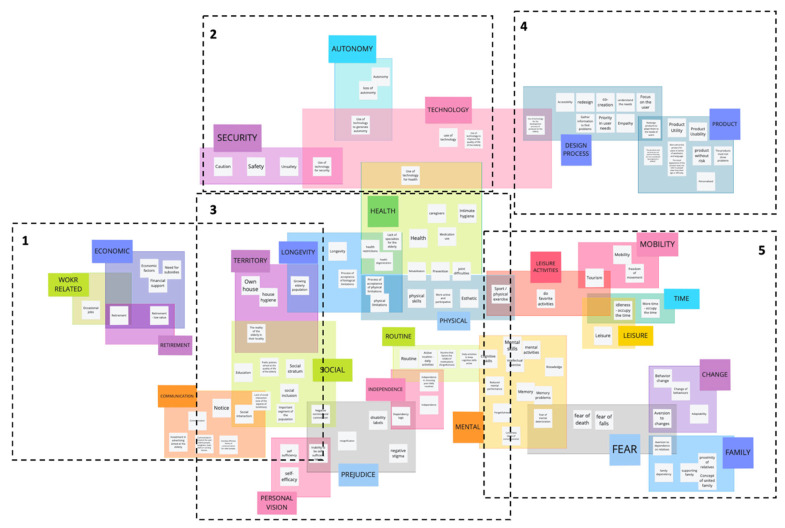
Mapping of activities with categories and subcategories. Source: own elaboration.

**Table 1 ijerph-20-04257-t001:** Key concepts found in each focus group.

Focus Groups				
Industrial Designers	Entrepreneurs	Health Professionals	Older Adults from Encampments	Self-Reliant Older Adults
74 concepts	173 concepts	233 concepts	145 concepts	138 concepts

(Source: prepared by the authors).

**Table 2 ijerph-20-04257-t002:** “Common concepts” category between the categories “Needs of older adults” and “Solutions for older adults”.

Common Concepts			
Accessibility Adaptability Retirement Retirement—low income The product’s visual appearance does not refer to people older than them or its difficulty. Warnings Daily activities to keep cognitive skills active Mental activities Self-efficacy Autonomy Self-sufficiency Aversion to dependence on relatives Aversion to change Co-creation Gathering information to find problems Communication User-directed communication: older adults, caregivers, municipalities, or nursing homes Concept of a united family Knowledge Negative socio-cultural connotation Personalized Caution Caregivers Health degeneration Dependence on family members Reduced mental performance	27. Developing effective forms of communication for older adults 28. Difficulties with joints 29. Empathy and understanding needs 30. Education 31. Sport/physical exercise 32. Forgetfulness 33. Aesthetics 34. Negative stigma 35. Social status 36. Intellectual exercise 37. Lack of older adult specialists 38. Lack of social interaction (as one aspect of loneliness) 39. Supportive family 40. Economic factors 41. User-centered 42. Cognitive skills 43. Physical skills 44. Mental skills 45. Household hygiene 46. Personal hygiene 47. Social inclusion 48. Independence 49. Independence in choosing their daily routines. 50. Insecurity 51. Social interaction 52. Investment in advertising aimed at older people 53. Leisure 54. Freedom of movement 55. Physical limitations 56. Longevity 57. More active and anticipatory 58. More time—occupies time 59. Fear of mental impairment 60. Fear of death	61. Fear of falling 62. Memory 63. Mobility 64. Own home 65. Change of behavior 66. Change of habits 67. Need for subsidies 68. Idleness—occupying one’s time 69. Loss of autonomy 70. Public policies aimed at quality of life for older adults. 71. Growing older adult population 72. Prevention 73. Prioritization of user needs 74. Memory problems 75. Process of acceptance of biological limitations 76. Process of acceptance of physical limitations 77. Product more attractive to users in terms of esthetics and language 78. Risk-free product 79. Products and services not used, so they are not considered “grandparents/older adults”. 80. Products should not show problems 81. Relatives nearby 82. Rehabilitation 83. Reality of older adults in their locality. 84. Perform favourite activities 85. Redesign products to suit users’ needs. 86. Redesign 87. Resignification	88. Health restrictions 89. Routine 90. Active routine—daily activities 91. Routine conducive to medication intake (forgetfulness). 92. Dependency labels 93. Disability labels 94. Inability to be self-sufficient labels. 95. Health 96. Significant population segment 97. Security 98. Loneliness (state of consciousness) 99. Financial support 100. Sporadic jobs 101. Tourism 102. Product usability 103. Use of technology to generate autonomy 104. Use of technology to improve the quality of life of older adults 105. Use of technology for product development process for older adults. 106. Use of technology for health 107. Use of technology for safety 108. Use of technology to improve the quality of life of older adults 109. Use of technology to improve the quality of life of older adults 110. Use of technology 111. Product usefulness

(Source: developed by the authors).

**Table 3 ijerph-20-04257-t003:** Relationships between frameworks and data collected.

	Basic Activities of Daily Living	Instrumental Activities of Daily Living	Enhanced/Advanced Activities of Daily Living
Transportation	Movement availability: the ability to control movements, and be able to move from one place to another with or without the support of a mobility aid or healthcare provider. Example: transferring (from bed, chair), walking, or support aids.	Daily mobility: the ability to move oneself within one’s neighbourhood and areas beyond this. Examples: travelling by car or public transportation.	Optimal mobility: relative ease and freedom of movement in all its forms is central to healthy aging. Examples: Travelling abroad, visiting other places, the feeling of freedom of movement.
Health	Basic Self-care: everything related to the basis of staying physically and mentally healthy. Examples: brushing your teeth, taking a shower, maintaining a regular sleep routine, eating healthy, and brain stimulation such as brain games.	Self-care: the ability to cope with illness and disability with or without the support of a healthcare provider. Example: seeking medical care when needed.	Advanced self-care: self-initiated behavior that people choose to incorporate to promote good health and general well-being. Examples: daily exercise to avoid fragility, habits to build a healthy life, annual physical examinations.
Living environments	Living space basics: spaces enabled to access basic daily activities. Examples: accessibility for bathing, eating, going to the toilet; inhabiting the space, being able to move around the space with or without the support of a healthcare provider.	Home activities: household activities undertaken to maintain their home; living space that enables other types of activities. Examples: housework, cooking, gardening, pet care, home maintenance, and repair; having spaces to do other activities at home such as working or hobbies.	Home Comfort: feeling free from stress or tension in the living space, due to the environmental conditions. Examples: the feeling of home, decoration, and renovation; the feeling of privacy.
Work and volunteer activities	-	Work-related tasks: duties or responsibilities that the individual performs in a job. They will vary according to the type of work. Examples: job-related, sales, ICT-related, and teamwork.	Complementary work-related tasks: non-mandatory activities that the individual enjoys doing in a job or volunteer activities. Examples: opportunities to learn, socialise with colleagues
Communication and social engagement	Communication skills: the abilities used when giving and receiving different kinds of information, such as listening, speaking, observing, and empathizing. They could also include handling ICTs. Examples: conversational skills, such as listening, empathizing, and managing interruption. ICT skills: using a mobile device such as a tablet or phone.	Social interaction: fulfilling the need of belonging to a larger social group, and feeling socially connected to family and friends. Examples: sharing with family and friends, having a support network, staying in touch by phone, or visiting people.	Social engagement: the extent an individual takes part in a broad range of social roles and relationships. Examples: participation in collective activities, church-going, volunteering, befriending neighbours, attending cultural events.
Leisure Activities	-	-	Leisure activities: activities engaged in for reasons as varied as relaxation, competition, or growth. Examples: hobbies, sports, entertainment, recreation, new learning opportunities, watching television, and listening to the radio.

(Source: developed by the authors).

**Table 4 ijerph-20-04257-t004:** Categorization of the concepts of needs and solutions using frameworks.

	Basic Activities of Daily Living	Instrumental Activities of Daily Living	Enhanced/Advanced Activities of Daily Living
Transportation	Movement availability: Mobility Physical limitations Loss of autonomy	Daily mobility: Adaptability Financial support	Optimal mobility: Tourism Autonomy Social status Independence Insecurity Freedom of movement
Health	Basic Self-care: Mental activities Routine Personal hygiene Routine that favors medication intake (forgetfulness) Medication use Daily activities to keep cognitive skills active Self-sufficiency Cognitive skills Physical skills Mental skills Memory Understanding needs Health Fear of mental impairment Reduced mental performance Forgetfulness Memory problems Health restrictions	Self-care: Rehabilitation Behavioral change Prioritization of user’s needs Acceptance process for biological limitations Process of acceptance of physical limitations Self-efficacy Risk-free product Caregiving Aversion to change Fear of death Fear of falling Degeneration of health Lack of older adult specialists	Advanced self-care: Intellectual exercise Change of habits Growing older population Adaptability Aesthetics Longevity Independence in the choice of daily routines Prevention Use of health technology
Living environments	Living space basics: Active routine—daily activities Need for subsidies Caution Products should not show problems Product usability Product usefulness Dependence on relatives	Home activities: Home hygiene Focus on the user Redesigning products to suit users’ needs	Home Comfort: Redesign Own home Co-creation Customized Aversion to dependence on relatives Public policies oriented to quality of life for older adults Use of technology to generate autonomy Use of technology to improve the quality of life of older adults Use of technology for the development process of products for older adults Use of technology for safety
Work and volunteer activities	-	Work-related tasks: Retirement Economic Factors Retirement—low value Knowledge Difficulties with joints	Complementary work-related tasks: Odd jobs Education Financial support
Communication and social engagement	Communication skills: Communication Use of technology Empathy	Social interaction: Social interaction Communication directed to the user: older adults, caregivers, municipalities, or nursing homes. Relatives nearby Supportive family Concept of a united family Lack of social interaction (one of the aspects of loneliness). Loneliness (state of consciousness)	Social engagement: Social inclusion The reality of older adults in their locality More active and participative Dependency labels Disability labels Inability to be self-sufficient labels Negative socio-cultural connotation Negative stigma Products and services not used, so that they are not considered “grandparents/older adults”. Significant population segment Resignification
Leisure Activities	-	-	Leisure activities: Sports/physical exercise Leisure Idleness—occupying time Do favorite activities More time—occupy time

(Source: developed by the authors).

**Table 5 ijerph-20-04257-t005:** Categories and subcategories extracted from the mapping.

Category	Subcategory	Description
Product design	- Design process - Product - Technology	Concepts related to product design to improve the lives of older adults.
Leisure activities	- Leisure activities - Leisure - Mobility - Time	Concepts related to time as a resource to be spent, linked to leisure activities.
Perception of older adulthood	- Fear - Change	Concepts associated with the aging phenomenon: getting older means changing, and these changes are accompanied by fears.
Context given by society	- Economic - Labor - Retirement	Concepts related to the societal context into which the older adult is inserted.
Independence in daily activities	- Economic - Labor - Retirement	Concepts related to the societal context into which the older adult is inserted.
Relationship with society	- Social - Communication - Personal Vision - Bias	How older adults relate to their social context, negative stigmas, and personal aspirations.
Necessary elements for the fulfillment of older adulthood	- Autonomy - Security	Concepts recognized as elements of concern for older adults to have a fulfilling life.
Well-being	- Health - Physical - Longevity	Concepts associated with the well-being of older adults. Some internal concepts must be addressed to improve well-being.
Context nearby	- Territory - Family	Concepts associated with the immediate context of the older adult, such as the place where they live and the people around them.

(Source: developed by the authors).

## Data Availability

The datasets used in this study are available from the corresponding author upon reasonable request.

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
