# Peer review of "Identifying the Needs of Older Adults Associated with Daily Activities: A Qualitative Study"

_ijerph, 2023, doi:10.3390/ijerph20054257_

Round 1
Reviewer 1 Report
Dear authors,
thank you very much for the opportunity to read your paper. I find the topic most important, and I share your concerns for understanding future needs in a society when the traditional demographic models have been overturned. I'm sympathetic to your take on forwarding older people's needs in a future society. However, I have problems in understanding your paper since the text lacks a basic level of precision:
1. what is the research problem that you try to address - including older people in design processes, understanding older people's needs during a design process, or giving a clear example of how to include people in design processes so that their needs are in focus for the design work?
2. what is your definition of products, services and design of products and services, since it seems to be more extensive than the traditional understanding of products and services, for instance architecture and the built environment is included in the introduction and as such latently included in the products and services. This is often a matter for discussion and it is recommended that the introduction also mentions the need for a more extensive understanding of products and services – which also the results suggests. This could also be a conclusion that when discussing needs of older persons, a holistic approach is necessary, which is both complicated and time-consuming. Thus, would your model create some advantages?
3. the title could be shortened to correspond better with what the study is focused on and tries to address.
4. the ending of the abstract is inconclusive – what do you mean by: contributing to a shared understanding of the design needs of self-supporting older adult users – as I take it, it must refer to the outcome of the co-creation initiative that you describe in your text.
5. validity and biases – one problem with ageing is that ageing is also cultural. Therefore, it would be of interest to state limitations of this co-creation models.
6. how many participants were involved in the study? This number is not indicated anywhere in the text, other than 6 focus groups with 4 to 10 participants, hence, 24 to 60 people? The appendix suggests 40 participants, with a clear dominance of women, why? And would this dominance influence the results?
7. how was the research realized considering what the researchers were doing and interacting with the participants, the methodology used? Some more explications are needed for the reader to grasp what a qualitative study with a phenomenological design is. What does didactic way means as a process, please describe it more clearly.
8. the section procedure seems to deal with ethical concerns rather than the procedure of the research. However, the GDPR is omitted – how was this legal framework addressed? Transcripts of the audio recordings was made – given the complexity of multiple voices this task seems impossible to achieve in a group with 10 persons – did any of the researchers participate making field notes?
9. results – why are the concepts only presented as a sum per group rather than listing them as concepts in words and then, perhaps, hihglighting reoccurring concepts? The study suggests that the 5 (how many focus groups were there, 6 or 5?) focus groups produced individual lists of concepts, but probably some must be reoccurring for all groups? Overall, the results section presents how the concepts obtained from the focus groups are sorted along the categories defined according the CREATE model, but this work was done by the researchers without the help of the focus groups or with the help of the focus groups? Something is missing here, why is this done? And what does this work produce? The assumption is that by sorting the many concepts according to the CREATE model it is possible to define according to which category a product or service intended for older people should be designed, a type of situating the needs in different fields of expertise?
10. table 4 is not completely in English.
11. discussion: the 251 concepts that are not addressed by solutions is a very unclear conclusion that the reader has severe problems of understanding.
Overall, the study seems to have been thoroughly performed but the final text omits important information that is needed to fully understand the study and how this study contribute to include older people in design processes of products or services. Rather, the study suggests that older people’s needs should be filtered through the CREATE model, but, why remains unclear.
Author Response
Author reply – Reviewer 1
We would first like to thank you for your highly pertinent comments, which have certainly helped improve our paper. We will answer each question and comment point by point, describing the changes that have been made.
- What is the research problem that you try to address - including older people in design processes, understanding older people's needs during a design process, or giving a clear example of how to include people in design processes so that their needs are in focus for the design work?
Thanks for this comment, we included the aim in lines 66-68 of the introduction: “Therefore, the aim of this paper is to identify the needs, directly related with ageing, that hinder the older adults’ ability to perform their everyday/daily activities. Focusing on self-sufficient older adults that live in Chile. “
- What is your definition of products, services and design of products and services, since it seems to be more extensive than the traditional understanding of products and services, for instance architecture and the built environment is included in the introduction and as such latently included in the products and services. This is often a matter for discussion, and it is recommended that the introduction also mentions the need for a more extensive understanding of products and services – which also the results suggests. This could also be a conclusion that when discussing needs of older persons, a holistic approach is necessary, which is both complicated and time-consuming. Thus, would your model create some advantages?
Thanks for this comment, we included the definition from Line 149 to line 153:
Products and services are defined for this research as products, devices, objects, and services as well as product-Service Systems. This concept is a combination of service and product that form a marketable set of products and services, jointly capable of fulfilling a client's need (Goedkoop,1999). For this research, the build environment, especially dwellings serve as the backdrop of most of the activities developed by older adults.
- The title could be shortened to correspond better with what the study is focused on and tries to address.
Thanks for your remark. We updated the title to one that captures the essence of the research: “Identification of needs of older adults associated with daily activities.: a qualitative study. “
- The ending of the abstract is inconclusive – what do you mean by: contributing to a shared understanding of the design needs of self-supporting older adult users – as I take it, it must refer to the outcome of the co-creation initiative that you describe in your text.
Thanks for the comment. We adapted the ending of the abstract. From line 33 to 35.
The resulting proposal is a way to situate the needs in different fields of expertise and thus be able to situate, broaden and deepen the identified map to share knowledge, between the user and key experts, oriented to co-create solutions.
- Validity and biases – one problem with ageing is that ageing is also cultural. Therefore, it would be of interest to state limitations of this co-creation models.
Thanks for the comment. We included the following statement in lines 328 -830: “This study must be understood in the socio-cultural context regarding the participants and the reality of the elderly in Chile. The methodology used to identify needs and solutions based on focus groups can be replicated with different groups.”
- How many participants were involved in the study? This number is not indicated anywhere in the text, other than 6 focus groups with 4 to 10 participants, hence, 24 to 60 people? The appendix suggests 40 participants, with a clear dominance of women, why? And would this dominance influence the results?
Thanks for your comment, we agree that its relevant to showcase the bias towards feminine participants. We included the following paragraph in lines 731 to 740:
“The focus groups were conducted in the cities of Santiago, Concepción and Talcahuano. The recruitment of the participants was approached through convenience sampling, from this perspective the gender variable was left open. A total of 16 subjects of the study were older adults, 9 industrial designers, 5entrepreneurs, and 10 health professionals working the elderly. Women represented 60% of the participants. This preponderance could be explained by the willingness to participate in social gatherings and on the culture that taints the everyday activities with a feminine bias. The same is true for caring activities and can explain why 9 out of 10 health professionals working with the elderly are women. There were 9 out of 10 men in the industrial designers’ group.”
- How was the research realized considering what the researchers were doing and interacting with the participants, the methodology used? Some more explications are needed for the reader to grasp what a qualitative study with a phenomenological design is. What does didactic way mean as a process, please describe it more clearly.
The research methodology is explained in 2. Methodology. The qualitative study is described in 2.1 lines 395 - 408:
A qualitative study, that involves collecting and analysing non-numerical data (e.g., text, video, or audio) to understand concepts, opinions, or experiences was done. The advantage of this type of research is that it can be used to gather in-depth insights into a problem or generate new ideas for research with a phenomenological design [29]. The data gathered through a qualitative study reflects the way that a person experiences or understands his or her world and showcases their particular worldview in relation to the question or problem proposed by the researcher. This research methos seeks to capture the perceptions and impressions of the participants, not to find universal truths. This was important, as co-creation relies on the experiences and the expertise of the users to design products and services, as well as considering the experience that other professionals have that interact with them (e.g., medical staff, among others) or that are considered in the design process (e.g., designers, entrepreneurs). The general methodological sequence used in the research is described in Figure 2.
Regarding the methodological approach and interaction between the researcher and the participants, this is described in lines 629 a 631: “in which the researcher acts as facilitator or moderator rather passive, opens the conversation and intervenes punctually, leaving the focus group participants to be the protagonists. The connection between the focus groups and the selection of phenomenology as a method was added, considering that the context of the study is co-creation.”
- The section procedure seems to deal with ethical concerns rather than the procedure of the research. However, the GDPR is omitted – how was this legal framework addressed? Transcripts of the audio recordings was made – given the complexity of multiple voices this task seems impossible to achieve in a group with 10 persons – did any of the researchers participate making field notes?
We included the following explanation in lines 593 - 496: “Before starting the focus group, the researcher and the participants read and signed the informed consent form. Then, a list of the participants of each focus group was generated, the audio of all the sessions was recorded, and later the transcription of each focus group was made. The anonymity of the subject was ensured in the transcription.”
- Results – why are the concepts only presented as a sum per group rather than listing them as concepts in words and then, perhaps, highlighting reoccurring concepts?
Table 1 shows the number of concepts identified in each of the focus groups. From this results, the tables were generated, and recurrent terms were identified. This is something that was done as part of the procedure but was not added to the article.
- The study suggests that the 5 (how many focus groups were there, 6 or 5?) focus groups produced individual lists of concepts, but probably some must be reoccurring for all groups?
Indeed, the final list (table 2) sums up the recurring concepts for all focus groups.
- Overall, the results section presents how the concepts obtained from the focus groups are sorted along the categories defined according the CREATE model, but this work was done by the researchers without the help of the focus groups or with the help of the focus groups?
After the focus group, during the analysis of the results, the researchers classified and grouped the needs and solutions mentioned.
- Something is missing here, why is this done? And what does this work produce? The assumption is that by sorting the many concepts according to the CREATE model it is possible to define according to which category a product or service intended for older people should be designed, a type of situating the needs in different fields of expertise?
Thank you very much for your comment. We have included from line 831 to 835
- table 4 is not completely in English.
Thank you very much for your comment. We have corrected table 4.
- discussion: the 251 concepts that are not addressed by solutions is a very unclear conclusion that the reader has severe problems of understanding.
Thank you for your comment. We have corrected the discussion, including limitations and future lines of work.
- Overall, the study seems to have been thoroughly performed but the final text omits important information that is needed to fully understand the study and how this study contribute to include older people in design processes of products or services. Rather, the study suggests that older people’s needs should be filtered through the CREATE model, but, why remains unclear.
Thank you for your comment. We included the following lines 621 to 628
Since the purpose is to design products and/or services for older adults, the process of classifying the identified needs is carried out using the CREATE model, which seeks to be incorporated since it classifies the activities and relates them to the human dimensions of the user, the socio-cultural factors of the environment, the physical environment and how these influence the particularities and characteristics of the older adult (demographic, psychographic, perceptual, cognitive, psychomotor), the use of technology and the need to solve tasks (complexity, familiarity, collaboration, time demands).

Reviewer 2 Report
Tittle: Needs and Product Design Solutions Analysis for Older Adults: A Qualitative Study
Version: January 24th, 2023
International Journal of Environmental Research and Public Health
General comment
This is a qualitative study with a phenomenological approach from three cities in Chile. The aim of this study was to analyze the needs of the Chilean elderly and their possible solutions through the design of products. By focus groups methodology several designers, entrepreneurs, health professionals and older people participants explored different topics related with the needs and solutions. Then of a categorization of concepts an affinity map to identify similarities and relationships between them is proposed. The topic of the manuscript is appropriate for the Journal. Identifying “Silver Economy” activities are imperative. It is of high interest to investigators and clinicians. However, paper has serious difficulties in all sections, especially in reference section. Several comments in discussion section should be revised prior to decide accepting for publishing. Paper should be revised and re-submitted. As consequence, major essential revisions are necessary. New review of a revised version is needed.
Major compulsory revisions
Before a new review, the references should be revised. This is the most difficult part of this paper. Non-inclusion of important references included in the main text is a problem. Why authors included alphabetic references. Please, let me show instructions for authors of International Journal of Environmental Research and Public Health
· References: References must be numbered in order of appearance in the text (including table captions and figure legends) and listed individually at the end of the manuscript. We recommend preparing the references with a bibliography software package, such as EndNote, ReferenceManager or Zotero to avoid typing mistakes and duplicated references. We encourage citations to data, computer code and other citable research material. If available online, you may use reference style 9.
Abstract non shows the aim of this paper.
Introduction section is too long, why included a paragraph of well-being and quality of life in general?. The aim of this paper is needs and solutions no well-being and quality of life.
Figure 1 is wrong about the classic classification of Activities of Daily Living used in gerontology. Please provide reasons. Please check:
· Costenoble A, Knoop V, Vermeiren S, Vella RA, Debain A, Rossi G, Bautmans I, Verté D, Gorus E, De Vriendt P. A Comprehensive Overview of Activities of Daily Living in Existing Frailty Instruments: A Systematic Literature Search. Gerontologist. 2021 Apr 3;61(3):e12-e22. doi: 10.1093/geront/gnz147. PMID: 31872238.
What is the conceptual framework for selected questions and topics in data collection techniques?. Please provide a reference to support it.
Difficulties with tables, please check Spanish words in table 5.
In discussion section each paragraph of comments and hypothesis should be reinforced by an adequate reference. For example page 11 line 299 or line 307, or line 317. This section should be review and re-write based in main findings. Many information provided is not included in the aim of this paper.
Finally, I wait for a revised version for an deeply analysis and review of this interesting paper.
Thanks for letting me review this manuscript.
This could be a nice paper.
Level of interest: An article whose findings are important to those with closely related research interests.
Quality of written English: Please check.
Statistical review: No.
Declaration of competing interests:
I declare that I have no competing interest.
Author Response
Author reply– Reviewer 2
We would first like to thank you for your highly pertinent comments, which have certainly helped improve our paper. We will answer each question and comment point by point, describing the changes that have been made.
- Before a new review, the references should be revised. This is the most difficult part of this paper. Non-inclusion of important references included in the main text is a problem.
Thank you very much for your comment. The references have been added to the final text, we regret the error and appreciate that it has been noted.
- Why authors included alphabetic references. Please, let me show instructions for authors of International Journal of Environmental Research and Public Health
Thank you very much for your comment. The references have been reformatted according to Chicago format as requested by the journal. We were confused with the "free formatting" section that appears in the instructions to authors.
- Abstract non shows the aim of this paper.
The objective of the work is included in the abstract: “The present study aimed to analyse the needs that impact the well-being of the Chilean elderly and their possible solutions through the design of products.”
- Introduction section is too long, why included a paragraph of well-being and quality of life in general? The aim of this paper is needs and solutions no well-being and quality of life.
Thank you very much for your comment. We understand the reviewer's point. However, meeting the needs and solutions for specific activities of older adults ultimately contributes to improving quality of life and well-being.
- Figure 1 is wrong about the classic classification of Activities of Daily Living used in gerontology. Please provide reasons. Please check:
Costenoble A, Knoop V, Vermeiren S, Vella RA, Debain A, Rossi G, Bautmans I, Verté D, Gorus E, De Vriendt P. A Comprehensive Overview of Activities of Daily Living in Existing Frailty Instruments: A Systematic Literature Search. Gerontologist. 2021 Apr 3;61(3):e12-e22. doi: 10.1093/geront/gnz147. PMID: 31872238.
Thank you very much for the valuable comment. Considering that products and services are acquired to meet user goals, such as work efficiency, leisure and health management, a good way to understand the classification of needs is the framework proposed by Czaja, (2019) to understand user needs and wants. Future work could contrast the current results, incorporate different frameworks for either needs or daily activities.
- What is the conceptual framework for selected questions and topics in data collection techniques? Please provide a reference to support it.
The conceptual structure was built based on the objectives and requirements of the research project.
- Difficulties with tables, please check Spanish words in table 5.
Thank you for your comment. The table has been corrected.
- In discussion section each paragraph of comments and hypothesis should be reinforced by an adequate reference. For example, page 11 line 299 or line 307, or line 317. This section should be review and re-write based in main findings. Many information provided is not included in the aim of this paper.
We regret the inconvenience; we had the references included but they were not incorporated in the final version of the manuscript. The error has been corrected.

Round 2
Reviewer 1 Report
Dear authors,
thank you for a paper that is well improved in its introductory parts but still lacks some rigour in discussion and conclusion. Over all the study is much clearer and well structured when it comes to scientific rigour, although some futher details could be added to improve its level of generalisability, ie. possible to repeat in another context.
I would like to make the following bullits:
page 2, sen 74: the focus on self-sufficient older adults that live in Chile limits the relevance of the study. Rather, it would be advisable to highlight and argue why the focus for the study is towards self-sufficient (odd choice of term, perhaps fully-abled living in ordinary housing). This makes the study less pertinent since needs are large with people with a long-term condition or dementia. To rephrase it, perhaps the ambition is to start co-creation processes with older persons, but this is the first step with the most obvious group since ethical considerations for this group are easier to meet. So, put the study in a context rather than limiting it to much. Furthermore, the following section 1.2 is very much about healthy ageing that includes both fully-abled and dis-abled older persons.
2. Reconsider the format of all figures considering legibility - figure 2 would benefit if the format was two rectangles broad and two rectangles high,
3. a comma is missing in sentence 403
4. table 2 - some type of coding that helps the reader to quickly understand the point of the table is required. I have severe difficulties in understanding this table.
5. figure 3 - would be beneficial with a larger format full side format if possible?
6. As I understand the study that aim was to test a co-creation process around older people's needs and that the contribution of the study is an organisational model. The discussion does not enter into detail here but for me as a reader this would be clear conclusion - how to organise a co-creational process among older people.
7. limitations, yes a limitation is that it is realised in a specific socio-cultural context, but presumably there are similarities with other contexts that would make the study relevant outside this context: most older people living in densely urban populated areas? One asset of the CREATE model could be that it allows for expanding outside this limitation - more argue the case, that yes this study is set to Chile, but mention what makes it relevant in another context - ageing is global as the introduction rightly points out.
8. future research could be to involve other groups of older people with emerging age-related problems.
9. conclusion: it is a very open conclusion, but the study suggests that it is the organisational structure that the CREATE methodology includes that makes it into a good instrument for starting co-creation processes regarding older people's needs.
Over all, the paper is much clearer and almost ready for publication. I would advise the authors to ponder more about the problem about qualitative research, i.e. this research is always depending on its methodology being clearly stated and part of its success is that a clear structure in the methods make the results probable to find in other contexts or in repeated studies. Looking forward to a revised version in which revision also includes the final parts of the paper.
Author Response
|
We would like to sincerely thank you for the valuable feedback that allowed us to improve the article. 1. page 2, sen 74: the focus on self-sufficient older adults that live in Chile limits the relevance of the study. Rather, it would be advisable to highlight and argue why the focus for the study is towards self-sufficient (odd choice of term, perhaps fully-abled living in ordinary housing). This makes the study less pertinent since needs are large with people with a long-term condition or |
|
dementia. To rephrase it, perhaps the ambition is to start co-creation processes with older persons, but this is the first step with the most obvious group since ethical considerations for this group are easier to meet. So, put the study in a context rather than limiting it to much. Furthermore, the following section 1.2 is very much about healthy ageing that includes both fully-abled and dis-abled older persons. Authors' response to comment 1: The reviewer is right. The project seeks to study product design with older adults, particularly participatory methodologies such as co-creation. It is in this sense that we limit ourselves at first to self-supporting older adults, as a first stage, and reduce the complexity both in the research process and in compliance with ethical considerations. It is expected that in a second stage we will be able to incorporate older adults with disabilities. 2. Reconsider the format of all figures considering legibility - figure 2 would benefit if the format was two rectangles broad and two rectangles high, Authors' response to comment 2: |
|
We have adjusted figures 1 and 2 to improve readability. 3. a comma is missing in sentence 403 Authors' response to comment 3: The reviewer is correct. We have included the comma. 4. table 2 - some type of coding that helps the reader to quickly understand the point of the table is required. I have severe difficulties in understanding this table. Authors' response to comment 4: We understand the reviewer's point, however the idea of the table is to be able to share the number of matching needs, without applying any a priori |
|
coding or filtering criteria. 5. figure 3 - would be beneficial with a larger format full side format if possible? Authors' response to comment 5: We agree with the suggestion. Unfortunately from an editorial point of view it is not possible. 6. As I understand the study that aim was to test a co-creation process around older people's needs |
|
and that the contribution of the study is an organisational model. The discussion does not enter into detail here but for me as a reader this would be clear conclusion - how to organise a co-creational process among older people. |
|
Authors' response to comment 6: The objective of the study is to generate an organizational model to be used as an entry point in co-creation processes with older adults to develop solutions, whether products or services. 7. limitations, yes a limitation is that it is realised in a specific socio-cultural context, but presumably there are similarities with other contexts that would make the study relevant outside this context: most older people living in densely urban populated areas? One asset of the CREATE model could be that it allows for expanding outside this limitation - more argue the case, that yes this study is set to Chile, but mention what makes it relevant in another context - ageing is global as the introduction rightly points out. |
|
Authors' response to comment 7: The reviewer is right and we have considered that argument and added in the limitations section. 8. future research could be to involve other groups of older people with emerging age-related problems. Authors' response to comment 8: We have incorporated your suggestion into future research. 9. conclusion: it is a very open conclusion, but the study suggests that it is the organisational structure that the CREATE methodology includes that makes it into a good instrument for starting co-creation processes regarding older people's needs. |
Authors' response to comment 9: We have incorporated your suggestion in the conclusions.

Reviewer 2 Report
The authors satisfactorily addressed my comments, however I recommended minor changes in figure 1. That I would like the authors to address regarding Activities of Daily Living (ADL) framework.
I continued with concerns about the reasons to insist in the classification as the same group basic ADL (Activities of Daily Living) and Instrumental ADL in figure 1 based in Czaja, Boot, Charness, & Rogers, 2019. The author´s answer is not enough to support interchangeable IADL tittle for two different areas of functional ability. Please I insist to read the reference recommended. Nowadays, in gerontology is clear the difference between basic and instrumental ADL. Figure 1 should be corrected.
Other comment, please check reference citation page 2 line 109 “client's need (Goedkoop,1999).” What is the reason to change citation form?
Authors should highlight in discussion section how is possible the generalization of their findings for other older communities
Author Response
We would like to sincerely thank you for the valuable feedback that allowed us to improve the article.
Comment1: The authors satisfactorily addressed my comments, however I recommended minor changes in figure 1. That I would like the authors to address regarding Activities of Daily Living (ADL) framework.
I continued with concerns about the reasons to insist in the classification as the same group basic ADL (Activities of Daily Living) and Instrumental ADL in figure 1 based in Czaja, Boot, Charness, & Rogers, 2019. The author´s answer is not enough to support interchangeable IADL tittle for two different areas of functional ability. Please I insist to read the reference recommended. Nowadays, in gerontology is clear the difference between basic and instrumental ADL. Figure 1 should be corrected.
Authors' response to comment 1:
We have taken into consideration the recommendation from the reviewer. We have adjusted our tables and figures to include B-ADL, I-ADL, and A-ADL, and have analyzed our data considering both articles. We have included as a reference the original article that is mentioned in regard to ADLs (De Vriendt et al.) cited by Constenoble, which was the paper suggested by the reviewer.
Comment 2. Other comment, please check reference citation page 2 line 109 “client's need (Goedkoop,1999).” What is the reason to change citation form?
Authors' response to comment 2:
The reviewer is right. Thank you very much for your comment. We have corrected it and put into format.
Comment 3. Authors should highlight in discussion section how is possible the generalization of their findings for other older communities.
Authors' response to comment 3:
The reviewer is right. In the discussion we have included their suggestions and considering that the theoretical models that compose the framework come from research done in the United States (Create) and Belgium (De Vriendt), we consider there is some universality in older people's needs and similar findings could be found if the study is replicated in other cultures.
